# The detection dogs test is more sensitive than real-time PCR in screening for SARS-CoV-2

Mohammed Hag-Ali [1]✉, Abdul Salam AlShamsi[2], Linda Boeijen [3], Yasser Mahmmod[1,4,7], Rashid Manzoor[1,7], Harry Rutten[3], Marshal M. Mweu[5], Mohamed El-Tholoth[1,6] & Abdullatif Alteraifi AlShamsi[1]

In January 2020, the coronavirus disease was declared, by the World Health Organization as a global public health emergency. Recommendations from the WHO COVID Emergency Committee continue to support strengthening COVID surveillance systems, including timely access to effective diagnostics. Questions were raised about the validity of considering the RT-PCR as the gold standard in COVID-19 diagnosis. It has been suggested that a variety of methods should be used to evaluate advocated tests. Dogs had been successfully trained and employed to detect diseases in humans. Here we show that upon training explosives detection dogs on sniffing COVID-19 odor in patients' sweat, those dogs were able to successfully screen out 3249 individuals who tested negative for the SARS-CoV-2, from a cohort of 3290 individuals. Additionally, using Bayesian analysis, the sensitivity of the K9 test was found to be superior to the RT-PCR test performed on nasal swabs from a cohort of 3134 persons. Given its high sensitivity, short turn-around-time, low cost, less invasiveness, and ease of application, the detection dogs test lends itself as a better alternative to the RT-PCR in screening for SARS-CoV-2 in asymptomatic individuals.

[1] Higher Colleges of Technology, Abu Dhabi, United Arab Emirates. [2] Federal Customs Authority, Abu Dhabi, United Arab Emirates. [3] DiagNose Netherlands B.V. and Four Winds K9 Solutions LLC UAE, Abu Dhabi, United Arab Emirates. [4] Department of Animal Medicine, Faculty of Veterinary Medicine, Zagazig University, Zagazig, Egypt. [5] School of Public Health, College of Health Sciences, University of Nairobi, Nairobi, Kenya. [6] Department of Virology, Faculty of Veterinary Medicine, Mansoura University, Mansoura, Egypt. [7] These authors contributed equally: Yasser Mahmmod, Rashid Manzoor. ✉email: hagali@hotmail.com

The ancient use of dogs for hunting, as well as their modern-time use for tracking, detecting bodies from disaster-struck areas, drugs, and explosives, stand as witness to the success and efficacy of their sense of smell[1].

The use of dog olfaction sense for the detection of disease is relatively new. In 1989, Williams and Pembroke were the first to report the dog's ability to detect malignant tumors[2]. Since then, many studies have been published advocating the ability of canines to detect many non-infectious and infectious diseases. Indeed, K9 dogs were successfully trained and used to detect different types of cancers[3], diabetes[4], epilepsy[5], bacteriuria[6], malaria[7], and viruses[8]. The chemical basis for such precise and sophisticated olfactory sense is the presence of volatile organic compounds (VOC) that represent a large range of stable chemical compounds, volatile at room temperature, and are detectable in body secretions. There is growing evidence that diseased cells produce a different pattern of VOCs than that of normal cells[9]. These VOCs are likely to have a distinct odor that seems unfathomable to humans.

Since the discovery of a novel coronavirus (the severe acute respiratory syndrome coronavirus 2, SARS-CoV-2) in Wuhan, China in late 2019, the virus has spread globally causing a pandemic. Soon after the start of the pandemic, detection of viral nucleic acid for screening purposes replaced the conventional thermal scanners around the globe due to the poor efficacy of thermal scanners. Although different viral nucleic acid (vRNA) detection methods have been reported[10,11], quantitative reverse transcription-polymerase chain reaction (RT-qPCR) is the one being widely used. The type of clinical specimen has been also shown to influence the sensitivity of RT-PCR. For example, lower respiratory tract specimens showed the highest positive rate followed by sputum and nasal swabs. Feces and blood had the poorest rates and no virus was detected in urine samples[12,13]. These tests vary in sensitivity and specificity. Pooled analysis from 16 studies estimated sensitivity of 87.8% (95% CI 81.5–92.2%) for an initial reverse-transcriptase PCR test[14]. Current available evidence also suggests that the specificity of the test is moderate (63–78%). It means that a positive test is highly suggestive of true COVID-19, but a negative test does not rule out the disease[15]. A number of factors, varying from reagents to the technical staff, have been proposed to be associated with the inconsistency of RT-PCR[16].

Moreover, high expenses are incurred in training staff, equipment, and reagents for continued and long-term mass screening is problematic. Therefore, search for alternate options, especially for screening purposes to cut the cost incurred is necessary.

There are currently 319 tests and sample collection devices that are authorized by the US Food and Drug Administration (FDA) for testing for COVID-19. These include 237 molecular tests and specimen collection devices, 69 antibody tests, and 13 antigen tests. (https://www.fda.gov/news-events/press-announcements/coronavirus-covid-19-update-january-19-2021).

RT-PCR is the most widely used test and we used it as the gold standard to evaluate the K9 test. Generally, tests for COVID-19 are judged almost exclusively on their sensitivity or how well they can detect viral proteins or RNA molecules in a specimen. This measure ignores the context in which the test is used. The contextual measure will look at how well infections will be detected in populations screened by repetitive use of the test. High-frequency testing using a low analytic sensitivity test would be superior in filtering COVID-19 cases than low-frequency testing with a high analytic sensitivity test[17].

The declaration by WHO, in March 2020, of COVID-19 as a pandemic sparked global multi-disciplinary approaches to deal with the disease that continued to claim thousands of lives daily worldwide. Several tests were developed for disease screening,

diagnosis, and monitoring. Those included molecular, serological, thermal, and other testing platforms. The tests would detect viral nucleic acid, viral antigens, antibodies, cytokines, or other viral-related indicators. Initially, thermal scanners were introduced at the ports of entry to control the spread of the disease. However, the identification of symptomless carriers significantly reduced the efficacy of their use in detecting COVID-19 positive patients. On the other hand, the widely used RT-PCR is costly, requires trained staff, expensive equipment, and reagents. The long turn-around-time hampers its use for fast and long-term mass screening. Keeping in view the success of canine olfaction in the detection of human disease with ample scientific evidence, it is suggested to evaluate the use of canines for the detection of COVID-19 patients.

Bayesian statistics is increasingly applied in different settings of clinical and health services research[18]. It is emerging as a good alternative to the classical frequentist statistical methods[19,20].

The assessment of the accuracy (sensitivity (Se) and specificity (Sp)) of COVID-19 diagnostic tests poses a challenge when the true infection status of humans cannot be determined owing to the absence of a perfect reference test. Furthermore, there are no available studies about Se and Sp of COVID-19 diagnostic tests and if it was available it is based on classical diagnostic evaluation. This approach is associated with much reporting bias[21,22]. The diagnostic test characteristics of real-time PCR and detection dogs tests have never been compared. On the other hand, Bayesian latent class analysis models (BLCMs) present a suitable option for the simultaneous estimation of sensitivity and specificity of two or more tests without any assumption about the underlying true disease status of each subject[23]. To the best of our knowledge, there are no published studies that apply a BLCM framework to quantify the accuracy of RT-PCR and K9 test for the assessment of COVID-19 infection in the human population. The objective of the present study is to estimate the Se and Sp of real-time PCR on nasal samples and K9 test on sweat samples for the detection of COVID-19 virus within a Bayesian framework and compare it to the traditional diagnostic test evaluation assuming PCR as a gold standard. Additionally, the true within-prevalence of COVID-19 infection in the investigated populations was estimated.

## Results and discussion

**Evaluating the performance of the K9 test to screen for SARS-CoV-2-infected persons using the RT-PCR as the gold standard.** The results of cross-tabulation between PCR as the gold standard and the K9 tests are presented in Table 1. The number of true positive, true negative, false positive, and false-negative cases are shown in cells A, D, B, and C, respectively.

Table 2 summarizes the indicators of diagnostic performance of dog detection and their interpretation.

The sensitivity (A/A + C*100) of dog detection was 83.3% (95% CI: 58.6–96.4%) and specificity (D/B + D*100) was 99.2% (95% CI: 98.8–99.5%) based on the numbers presented in Table 1. At a prior probability of infection of 0.01, the positive predictive

**Table 1 Cross-tabulation between RT-PCR and the K9 dogs olfaction tests.**

| Test | Real-time PCR test | | Total |
|---|---|---|---|
| | Positive | Negative | |
| K9 dogs olfaction test | | | |
| Positive | 15 (A) | 26 (B) | 41 (A + B) |
| Negative | 3 (C) | 3246 (D) | 3249 (C + D) |
| Total | 18 (A + C) | 3272 (B + D) | 3290 (A + B + C + D) |

**Table 2 Calculations of Statistical Indicators and their Interpretation.**

| Statistical Indicator | Value | Interpretation |
|---|---|---|
| Sensitivity | 83.3% | Sensitivity is low which means that a positive test rarely occurs in those with the COVID virus |
| Specificity | 99.2% | Specificity is high meaning that a negative test often occurs in those without COVID-19 |
| Positive predictive value (PPV) | 52% | PPV is very low meaning false positives are common |
| Negative predictive value (NPV) | 99.8% | NPV is very high meaning that the test is useful for screening |
| Prevalence | 0.005 | The prevalence of COVID is very low (about 0.5%), hence prone to false positives |
| Accuracy | 99.1% | Proportion correctly classified |

**Table 3 Summary of the positive and negative predictive values for the detection dogs test for a range of prior probabilities of infection of COVID-19.**

| Prior probability of infection infection | Positive Predictive Value | Negative Predictive Value |
|---|---|---|
| 0.005 | 0.35 | 1 |
| 0.01 | 0.52 | 1 |
| 0.05 | 0.85 | 0.99 |
| 0.1 | 0.92 | 0.98 |

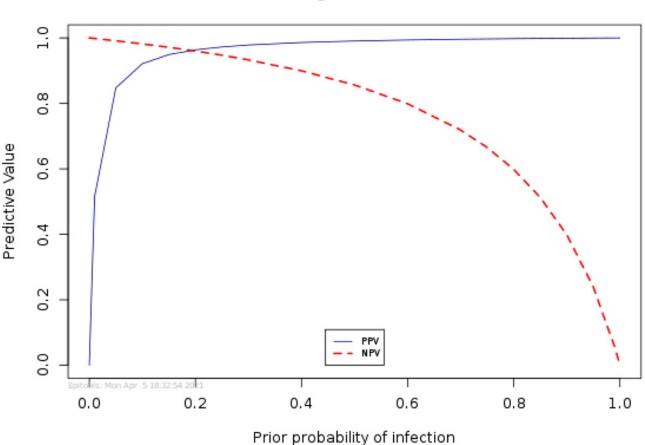

**Fig. 1 Positive and negative predictive values for the detection dogs' olfaction test (K9 test) using RT-PCR test as the gold standard.** There is a reciprocal variability relationship between PPV and NPV of K9 test at different prior probabilities of infection.

value of dog detection is 52%, which reflects the probability that individuals with a positive screening test (dog detection) do truly have the disease (COVID-19 positive). The negative predictive value was 99.8%, which means the probability that individuals with a negative screening test (dog detection) truly do not have the disease (COVID-19 negative)[24–26].

The accuracy (proportion of individuals correctly identified by detection dogs (true positives and true negatives)) was 99.1%. On the other hand, the proportion of individuals incorrectly tagged by detection dogs was 0.9%.

Table 3 summarizes the positive and negative predictive values for the dog detection for a range of prior probabilities of infection of COVID-19. The presented data show that as the prior probability, which reflects the prevalence of infection, increases, the PPV of the test will also increase. A 10 times increase in prevalence, from 0.005 to 0.05, will more than double the PPV of the test from 0.35 to 0.85. Similarly, a 10 times increase in prevalence from 0.01 to 0.1, will almost double the PPV of the test, from 0.52 to 0.92. The NPV, however, shows a slight decrease as the prevalence increases. This reciprocal variability relationship between PPV and NPV at variable prior probabilities of infection is illustrated in Fig. 1. The high NPV of the K9 dogs test supports the use of this test for screening purposes as it will screen out individuals who do not need to do the more expensive, time-consuming, and labor-intensive RT-PCR test.

**Comparing the diagnostic accuracy of the K9 and the real-time PCR tests using Bayesian latent class analysis (LCA).** Out of the total 3290 observations, 156 observations were excluded from the Bayesian analysis due to missing information or subjects not being from Africa or Asia. The analysis was carried out on the clean data with complete observations that included 3134 observations.

The two subpopulations from Africa and Asia yielded 7 degrees of freedom (df) sufficient to estimate 6 parameters (sensitivity and specificity of the two tests, and two subpopulations prevalences). To the best of the authors' knowledge, there is no available literature/information on the diagnostic Se and Sp estimates for COVID-19 diagnostic tests. Thus, non-informative priors (beta (1, 1)) were used to fit the Bayesian model since no reliable prior information was available for any of the aforementioned parameters. Notably, the hypothesis: $H_0$: p.se, p.sp = 0, was evaluated using a Bayesian

**Table 4 Cross-tabulated results for combinations of two diagnostic tests K9 dogs and RT_PCR for diagnosis of COVID-19 in the human population ($n = 3134$) in UAE.**

| Population based on location | Tests combinations (Test 1: K9; Test 2: RT-PCR) | | | | Total |
|---|---|---|---|---|---|
| | $++$ | $+-$ | $-+$ | $--$ | |
| Asia (pop 1) | 12 | 18 | 3 | 2895 | 2928 |
| Africa (pop 2) | 3 | 0 | 0 | 203 | 206 |
| Total | 15 | 18 | 3 | 3098 | 3134 |

$P$-value. The goodness-of-fit of the Bayesian model was evaluated using the posterior predictive $P$-value.

The model was initialized with two Markov Chain Monte Carlo chains with different values. Each chain comprised 600,000 samples, with the first 200,000 being discarded as the burn-in. Convergence of the MCMC chain was assessed by visual inspection of the time series plots of selected variables as well as by inspecting Gelman–Rubin diagnostic plots, density plots, and autocorrelation plots using three sample chains with different initial values[22]. The posterior distribution of the population prevalences, the sensitivity, and specificity of the two tests, as well as the conditional covariances, were reported as the median and the corresponding 95% posterior credibility intervals (PCI).

Results of cross-tabulation (contingency) of the dichotomous outcome of K9 dogs and RT-PCR for detection of SARS-CoV-2 are shown in Table 4. The estimates of posterior median and 95% PCI of true prevalence and Se and Sp of the two tests are shown in Table 5.

**Table 5 Test estimates of two screening tests K9 and RT_PCR for diagnosis of COVID-19 in the human population ($n = 3134$) in UAE.**

| Parameter | Tests estimates (test 1: K9; test 2: RT-PCR) | |
|---|---|---|
| | Median | 95% PCI |
| Prevalence in Asia (pop 1) | 0.02 | 0.006–0.043 |
| Prevalence in Africa (pop 2) | 0.007 | 0.003–0.014 |
| Se of K9 dogs | 0.89 | 0.65–0.99 |
| Se of RT-PCR | 0.73 | 0.38–0.99 |
| Sp of K9 dogs | 0.99 | 0.99–1.0 |
| Sp of RT-PCR | 0.99 | 0.99–1.0 |
| p.se | 1 | 0.00–1.00 |
| p.sp | 0 | 0.00–1.00 |

Se of K9 dogs 89% (95% PCI: 65–99%) was higher than Se of RT-PCR 73% (95% PCI: 38–99%) but Sp of K9 dogs 99% (95% PCI: 99–100%) was comparable to the Sp of RT-PCR 99% (95% PCI: 99–100%). The estimated true prevalence of COVID-19 in the Asian population of 0.02 (95% PCI: 0.006–0.043) is higher than the estimated prevalence in the population of Africa 0.007 % (95% PCI: 0.003–0.014) (Table 5). In the model, the parameter, p. Sp and p.Se of both tests were not significant, i.e., the 95% PCI cover 0.

Our findings showed that both tests succeeded in detecting SARS-CoV-2 in nasal swab samples (RT-PCR) and in armpit sweat samples (K9 test). The K9 test showed a higher Se than PCR, but the two tests showed a high and similar Sp (99%). The difference in the Se between both tests can be argued by many factors. For instance, (a) the type and size of the sample submitted for the diagnostic test (b) the sampling technique, and procedures of sample preparations, and (c) load of the virus in the sample. Sethuraman et al.[27] found that the accuracy of RT-PCR for detection of COVID-19 is likely to vary depending on the stage of the disease. Moreover, Wölfel et al.[28] added that the degree of viral multiplication or clearance from the body may have an impact on the RT-PCR performance. The high Sp of both tests indicates that they have a great potential to correctly identify the truly negative COVID-19 individuals. This is the first study to estimate the diagnostic performance of the K9 test and RT-PCR for detection of SARS-CoV-2 using Bayesian LCA.

Our findings for the Se and Sp estimates of RT-PCR are in agreement with previous studies. For instance, Ren et al.[29] reported that Se (78%) and Sp (99%) for RT-PCR. On the other hand, Wang et al.[13] reported a lower Se of RT-PCR (63%) in comparison to our study results. However, Watson et al.[30] found that the Se of RT-PCR is varying according to the type, quality, and site of the sampling. Wang et al.[13] found that the Se of RT-PCR in broncho-alveolar lavage was 93%, while it was 72% for sputum and only 32% for throat swabs. Other studies by Xie et al.[31] and Yu et al.[32] reported a similar Sp estimate (100%) of the RT-PCR method to our study estimates.

Bayesian LCA model requires fulfillment of a number of model assumptions and conditions. The first assumption of the LCA model is that the two diagnostic assays under evaluation (RT-PCR and K9 test) are conditionally independent given disease status. This assumption is certainly achieved because RT-PCR and K9 dogs are based on two distinct biological identification mechanisms. The second assumption that the test characteristics (Se and Sp) should be constant across the study populations was fulfilled because geographical location (stratifier) would not affect test characteristics. The final assumption is that the prevalence of infection/disease status should differ between the populations. That assumption was verified, as posterior estimates of prevalences in our study populations were markedly different (Table 5).

The data on K9 and RT-PCR tests of 3134 persons (Africans and Asians only) were subjected to the LCA modeling to estimate the sensitivity and specificity of PCR and K9 tests for diagnosis of COVID-19 infection without the assumption of an existing reference standard. K9 test has a higher sensitivity than the RT-PCR test. The median estimate of specificity of both tests was high and comparable at 99%.

Recently, a study identified a unique profile of VOCs in sweat samples for the diagnosis and staging of adenocarcinomas. In fact, it is suggested that sweat can represent substances in the blood, but owes an advantage of being less complex in chemistry than the blood[33]. The results in the present study also suggest the presence of a unique profile of VOCs in sweat samples of COVID-19 patients being detected by K9 dogs. In a subset of the 3290 subjects screened, 13 showed symptoms of cough and flu. All those 13 tested negative by RT-PCR as well as the K9 dog test. This supports the notion of a distinctive COVID-19 odor. Therefore, a study is being planned to investigate the sweat VOC profiles from COVID-19 patients.

In our study, we used two different diagnostic test evaluation methods, Bayesian modeling and gold standards for the estimation of the K9 dog's performance in the detection of SARS-CoV-2. However, from the application point of view, the Bayesian modeling is preferable over the conventional statistical modeling using PCR as the gold standard for many reasons. For instance, BLCA estimates the diagnostic accuracy of tests and true prevalence when the gold standards are imperfect. Moreover, it enables us to show the superiority of a novel test over an old test, even if the old test is the imperfect gold standard as it was shown in many previous studies, which used BLCA for estimating the diagnostic performance of several infectious disease diagnostic[34–37].

In summary, the results of detection of COVID-19 infection in the sweat samples using the K9 test and RT-PCR were available for 3290 persons in UAE, the majority representing two main populations (Africa and Asia).

The specificity of the detection dog test (K9 test) was very high (98.2%) indicating that the dog detection test has a great performance in correctly identifying and detecting those individuals who are truly negative for COVID-19. Test validation studies assuming perfect reference tests are common, but with the potential to introduce bias in the estimation of index test(s) performances[38]. Paradoxically, the high Se of PCR methods compared with other assays makes it almost impossible to define a proper 'gold standard' for diagnostic performance evaluation. The high negative predictive value (NPV) of the dog detection test is a strong indicator that the test can be implemented as a screening test for COVID-19.

We concluded that diagnostic performance of detection dogs (K9) test lends itself as a quick routine screening test of COVID-19 and it poses as a preferable candidate for its merits such as very low cost, less required facilities and resources in addition to low risk of COVID-19 transmission in comparison to PCR during sample collection.

## Methods

**Study population.** The study population consisted of 3290 persons attending the COVID-19 screening center in a migrant laborers residential area in Abu Dhabi, UAE. All the attendees to the center are males, between 19 and 67 years of age with 60% of the subjects in the 26–40 years old range. More than 90% of the subjects are from Asia, with the largest representation (66%) from India and Pakistan (38% and 28%, respectively). All the screening center attendees were visiting the center to do their mandatory RT-PCR COVID-19 screening test. They were asked to donate an

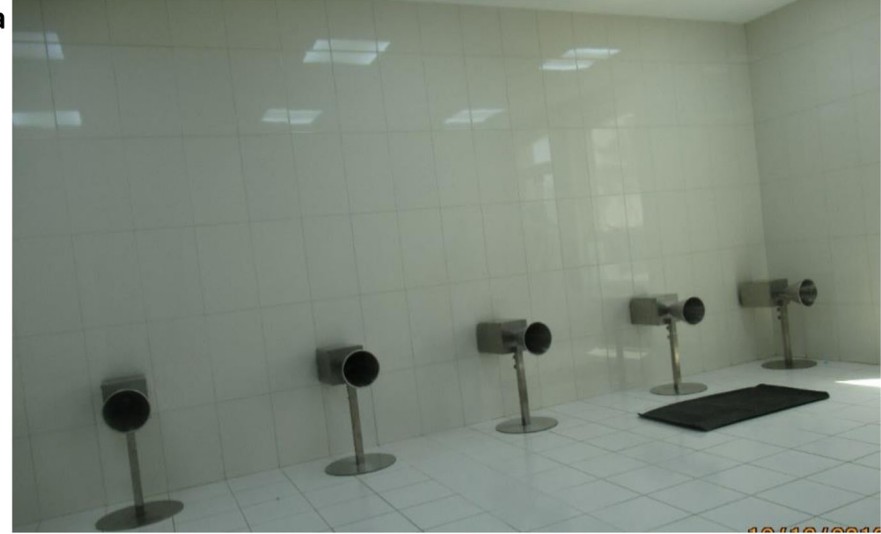
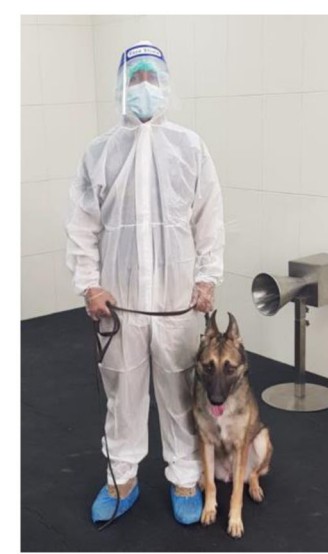

**Fig. 2 Training stations lined up at the Federal Customs Authority (FCA) facilities in Abu Dhabi, United Arab Emirates, for the training of explosives detection dogs. a** Training devices with samples' containers placed right behind the metal olfaction cones. **b** The dog handler is deployed to work with the dog as a team.

armpit sweat sample. Those who agreed were asked to provide their written consent. The RT-PCR specimen was collected and processed by a healthcare provider. The result of the PCR test was not known at the time of collecting and testing the sweat sample by the detection dogs. The specimens collected were barcoded early on before testing.

**Sample collection**. Two types of samples were collected from different persons; one for training the dogs and the other was for the comparative studies.

*Samples for training the dogs to recognize and imprint the SARS-CoV-2 odor*. Sweat samples ($n = 234$) were collected from armpits of COVID-19 patients using a sterile gauze pad at a hospital in Abu Dhabi, UAE. These patients showed respiratory manifestations and were identified by reverse transcription-polymerase chain reaction (RT-PCR) as COVID-19 positive using nasal swabs. Negative control sweat samples ($n = 144$) were collected from the same hospital, from persons that did not show any recent respiratory manifestations and their nasal swabs tested negative for SARS-CoV-2 by RT-PCR. The collected samples were not screened for other coronaviruses infecting humans. After collection, the samples were barcoded and transported in sterile collection bottles to the dogs training center.

The armpit sweat was selected in our study because it contains volatile organic compounds at the normal body temperature, and it is the primary source of the body odor making it more suitable for sniffing by the detection dogs[39]. Moreover, sweat is not a route for SARS-CoV-2 excretion from infected patients[40], rendering the sample to be non-infectious[41].

The presentation of the samples, either positive or negative, to the dogs was carried out via a sample container attached to a metal cone (Fig. 2a). The dog handler is deployed to work with the dog as a team (Fig. 2b). The K9 detection dogs are trained to sit in front of the metal cone attached to the container with SARS-CoV-2-positive samples.

*Samples for evaluation of trained detection dogs clinical performance in COVID-19 screening*. A total of 3290-armpit sweat samples were collected from individuals visiting the COVID-19 screening center in Mafraq Workers City, Abu Dhabi, UAE. One sweat sample was collected per participant; no-repeat samples. The samples were subjected to screening using trained K9 dogs. Nasal swabs from the same individuals involved in our study were tested by SARS-CoV-2-RT-PCR protocol to identify COVID-19 positive and negative patients.

Samples that were collected from the hospital were transported on the same day to the Federal Customs K9 facility.

The positive and negative samples were transported separately in medical coolers with ice packs but under the same conditions. From there, they were stored in the fridge and mostly used the same day. All samples were taken out of the fridge to come to "room temperature" before presented to the dogs. If samples were used a day later, they were stored in an air-tight container in the fridge.

The operational samples were collected by the participant, under the supervision of an assigned nurse or medical doctor explaining the procedure and making sure that the sampling protocol was followed. The samples were stored in air-tight containers.

**Training of dogs**. Four male Federal Customs Explosive detection dogs (K9) were selected for the training to screen for SARS-CoV-2 using axillary sweat samples. Dogs were fully trained and certified twice for explosive detection. The dog teams have been working together for over two years as a team (dog and handler). We selected experienced explosive detection dogs (K9) and their handlers because the dogs already had have an excellent search pattern and they understand how to indicate the scent of a variety of explosives. No experiments were performed on any animals. The welfare and safety of the working dogs were ensured as per guidelines in the Dutch Animal welfare regulations (https://www.government.nl/topics/animal-welfare/welfare-of-pets).

Briefly, training of dogs was done in three stages, i.e., (i) training search pattern on search stands, (ii) imprinting, and (iii) operational training and testing stage. The first stage of training on search patterns on search stands was completed in 1 day by dog trainers. The imprinting stage was completed in four phases. Training in imprinting phase one was completed in 2 days containing an average of three sessions per day, consisting of four line-ups on average; and a total of 20 positive samples were used. Training in imprinting phase two was completed in 2 days containing an average of three sessions per day, consisting of four line-ups on average, and a total of 24 positive samples and 192 mock samples. Training in imprinting phase three was completed in 1 week, 5 days training consisting of three training sessions per day, where each session contained an average of four line-ups. Training in imprinting phase four was completed in 4 weeks, 5 days per week training, containing an average of three sessions per day; each training session consisted of an average of four line-ups, and a total of 234 positive samples and 144 negatives samples were used in the imprinting phase three and four. After completing the imprinting phases, the trained dogs were taken to the operational setting for acclimatization and testing. During this stage, the samples were collected and tested from randomly selected persons. Usually, there was one trainer, one data registrar, and one team leader present during imprinting, training, and operational training and testing sessions. Sometimes, other instructors or the authors of the paper were also present for monitoring purposes.

The details of the K9 training are available in the Supplementary information file.

**Quantitative reverse transcription-polymerase chain reaction for SARS-CoV-2 detection**. Nasal swabs from the same individuals involved in our study were tested for SARS-CoV-2 by qRT-PCR protocol developed by the Chinese Center for Disease Control and Prevention[42] and recommended by WHO referral laboratories[43]. QIAamp Viral RNA Mini Kit (Qiagen, Germany) was used for RNA extraction based on the recommended instructions by the manufacturer. The TaqMan probe and primers and for RT-PCR test were synthesized by Sigma Chemical Company, USA. Each qRT-PCR reaction mix contained 1× TaqMan Fast Virus-1-Step Master Mix (Life Technologies), 250 nM of TaqMan probe, 400 nM of each PCR primer, and 5 µL of extracted RNA. qRT-PCR was performed with a Thermal Cycler (BioRad, Model CFD3240). The RT-PCR had a temperature profile of 55 °C for 20 min, then 95 °C for 3 min, followed by 45 cycles of amplification (95 °C for 30 s and 60 °C for 1 min).

**Bayesian latent class analysis**. Owing to the imperfection of reference tests, the Bayesian latent class analysis models (BLCMs) present a suitable option for the

simultaneous estimation of sensitivity and specificity of two or more tests without any assumption about the underlying true disease status of each subject[23]. Basically, BLCMs are premised on three key assumptions: (1) the target population should consist of two or more subpopulations with different prevalences, (2) the sensitivity and specificity of the index tests should be constant across the subpopulations, and (3) the tests should be conditionally independent (CID) given the disease status[23]. BLCA provides a solution when no ideal reference method of diagnosis exists. This statistical approach assumes that a person can be classified into one of two infection statuses (i.e., infected or not infected), while the true infection status is unknown.

### Statistical analysis

*Evaluating the performance of the detection dogs test to screen for SARS-CoV-2-infected persons using the RT-PCR as the gold standard.* At first, the collected data were screened for unlikely or missing values before running any valid statistical analysis. No data were excluded on this basis. Subsequently, a descriptive statistical analysis was performed for both test results of PCR and olfaction tests. Calculation of true positive (TP), true negative (TN), false positive (FP), and false-negative (FN) was conducted. The diagnostic sensitivity, diagnostic specificity, positive predictive values (PPV), and negative predictive values (NPV) were estimated. The PPV and NPV have been calculated at a different prior probability of infection.

*Comparing the diagnostic accuracy of the detection dogs and the real-time PCR tests using Bayesian latent class analysis (LCA).* A Bayesian latent class model fitted in OpenBUGS v3.2.2 was used to infer the sensitivity and specificity of the two tests (K9 test and RT_PCR)[44], as well as the two populations (African and Asian) prevalences as per the standards for reporting diagnostic accuracy studies that use STARD-BLCMs[45]. In particular, the test accuracy of the diagnostic tests under evaluation was assumed to be similar across the study populations, i.e., sensitivity and specificity constancy. The two tests under evaluation (K9 test and RT_PCR) are based on two different principles of action assuming a conditional independence between the two tests. Thus, we implemented two tests (conditionally independent) in two populations BLCA model[46].

We divided our study population into two subpopulations of humans with different densities according to their country of origin (nationality) and geographical location, which were perceived to have different true prevalences of COVID-19 infection. Additionally, we assumed that nationality would not influence Se and Sp. Population 1 refers to people originating from Asian countries. While population 2 refers to people originating from African countries.

**Reporting summary**. Further information on research design is available in the Nature Research Reporting Summary linked to this article.

## Data availability

The data sets generated and analyzed during the current study are available in the figshare repository[47] https://doi.org/10.6084/m9.figshare.13289777. All other data are available from the corresponding author on reasonable request.

## Code availability

A Bayesian latent class model fitted in OpenBUGS v3.2.2 using R software was used. The BLCA model was implemented in R version 3.5.1 using (BRugs) package. The R Script, as well as the run printout showing the Density and Trace plots are included in Supplementary Data 2 and the Supplementary information files, respectively. There is no restriction in running the R software.

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

## Acknowledgements
We thank the Federal Customs Authority K9 Unit for allowing them to use their facilities.

## Author contributions
All authors participated in several planning sessions. M.H.-A. analyzed the data, wrote the manuscript, played important roles in designing the human study protocols, supervised the human element of the study, and coordinated the human-animal parts of the study. A.S.A. obtained animal use permission, carried out the animal study, trained the dogs, played important roles in designing the animal and human study protocols, obtained consent approval, supervised the human element of the study. L.B. obtained animal use permission, carried out the animal study, trained the dogs, played important roles in designing the animal study protocols, supervised the animal element of the study. Y.M. analyzed the data, wrote the manuscript, and played important roles in designing the human study protocols. H.R. obtained animal use permission, carried out the animal study, trained the dogs, played important roles in designing the animal study protocols. R.M. analyzed the data, wrote the manuscript, and played important roles in designing the human study protocols. M.M.M. analyzed the data, wrote the manuscript. M.E.-T. analyzed the data, wrote the manuscript, and played important roles in designing the human study protocols. A.A.A. analyzed the data, wrote the manuscript, played important roles in designing the animal and human study protocols, obtained consent approval, and supervised the human element of the study. Y.M. and R.M. contributed equally to this work. M.H.-A., A.S.A., L.B., and A.A.A. are the guarantors of this work and, as such, had full access to all the data in the study and take responsibility for the integrity of the data and the accuracy of the data analysis.

## Competing interests
The authors declare no competing interests.
