## [Peer Review File · Communications Biology]

Reviewers' Comments:

Reviewer #1:

Remarks to the Author:

Thank you sending me the manuscript entitled "Detection Dogs Test is more sensitive than Real-Time PCR in Screening for SARS-CoV-2" for review. This work describes a comparison between two diagnostic methods for COVID 19; PCR and detection dogs. With a specific statistical Model, reliability of the test were evaluated without having a true gold standard. The authors concluded that detection dogs are suitable methods for screening COVID 19 infections, even better than PCR testing.

A proof of concept that dogs are able to identify COVID 19 in saliva and Tracheobroncheal smears were published by Jendry et al. (Jendry et al. BMC Infectious Diseases (2020) 20:536

<https://doi.org/10.1186/s12879-020-05281-3>). In this study sweat was a novel specimen.

For the scientific community of medical detection dogs, major information on training, dogs, and testing are missing.

It is of utter importance to critically evaluate the odor samples that are presented to the dogs, to judge, what the dogs possibly identify.

The author found a very high specificity for the detection dogs. For a screening test for Covid 19 a high predictive value. With the relative low prevalence of the infection, the NPV is nearly perfect. So if the authors could describe the dog work in more detail, it would be great to spread this information.

Reviewer #2:

Remarks to the Author:

The study is important and timely. However in its current form the manuscript does not allow the reviewer to fully appraise the scientific rigour of the study nor the reader to fully understand the conclusions reached.

From the abstract it is not clear how the paper can conclude the the dog is more successful than the PCR screening. As it is unclear what gold standard the two are being compared to? This is an issue throughout for me as the first analysis assumes the PCR is the gold standard, yet the Bayesian analysis concludes otherwise, which seems a cyclical argument and is not coherently dealt with in the manuscript

For example..

Line 121.. in the paragraph above the authors describe that they use BLCA as the two test may produce different results and we cannot make assumptions which represents the true disease state yet here they present TP, TN FP etc.. how can these be derived, if the true gold standard is unknown? Similarly section 150 – 168 all assume the PCR is the gold standard? This discrepancy between the two types of analysis is implicit throughout the paper and requires more synthesis and clearer discussion.

The background section is extremely short and goes from describing the multiplicity of uses of dogs to stating the specificity of PCR for Covid-19. It requires some introduction to Covid-19, the needs for screening and references to the sensitivity of PCR for detecting COVID-19 from different sample types.

The sections on training and testing of the dogs give inadequate detail to allow full appraisal of the study. Was training blind? Was testing blind or double blind? Who laid out the samples? Who handed the dogs? How many dogs were they? What breeds were they? How many times was each sample presented? What was the ratio of positive to negatives presented to the dogs? How were the dogs rewarded? There are sound examples of the sort of details required when reporting dog

detection studies in References 9-13 in the current paper, and also studies describing the multiple pitfalls that need to be avoided when training dogs (e.g. Edwards et al 2017; Guest et al 2020)

Edwards, T.L., Browne, C.M., Schoon, A., Cox, C., Poling, A. (2017). Animal Olfactory Detection of Human Diseases: Guidelines and Systematic Review, *Journal of Veterinary Behavior: Clinical Applications* doi: 10.1016/j.jveb.2017.05.002

Guest CM, Harris R, Anjum I, Concha AR and Rooney NJ (2020) A Lesson in Standardization – Subtle Aspects of the Processing of Samples Can Greatly Affect Dogs' Learning. *Front. Vet. Sci.* 7:525. doi: 10.3389/fvets.2020.00525

Lines 71... was processing of the positive and negative samples the same? What happened if a patient showed no symptoms and tested positive or showed symptoms and tested negative? Were they eliminated from both groups?

Line 78 should read arm pit not bit

Line 81 should read non-infectious not infective

Line 92 there appears to be text missing as a sentence finishes mid phrase.

In line 115 the authors describe this to be the first application of BLCA to Covid -19 dog detection data.. but I question, with only 8 positive cases according to PCR, is this an appropriate sample and analysis technique. Has the analysis method been used on other models of canine detection and if so to what benefit? What's more the authors need to expand the discussion to explain what this analysis brings to the debate and describe fully whether their Bayesian or more conventional NPV and PPV values are likely more accurate

Table 3 presents analysis which have not been introduced nor fully described in the methods

The logic of splitting the sample into Asian and African is unclear, and since there are no positive cases in the African subpopulation, it seems this adds little to the paper

Reference 10 the formatting is incorrect

Cover Letter

Response to Reviewers

of the Manuscript titled

“The Detection Dogs Test is more sensitive than Real-Time PCR in screening for SARS-CoV-2”

By

Mohammed Hag-Ali,¹ Abdel Salam Alshamsi,² Linda Boeijen,³ Yasser Mahmmod,^{1,4} Rashid Manzoor,¹ Harry Rutten,³ Marshal M. Mweu,⁵ Mohamed Eltholoth,^{1,6} Abdullatif Alshamsi¹

Authors Response to Reviewers' Remarks to the Authors

Reviewer #1 (Remarks to the Author):

Thank you sending me the manuscript entitled " Detection Dogs Test is more sensitive than Real-Time PCR in Screening for SARS-CoV-2" for review.

1. Reviewer 1: This work describes a comparison between two diagnostic methods for COVID 19; PCR and detection dogs. With a specific statistical Model, reliability of the test were evaluated without having a true gold standard. The authors concluded that detection dogs are suitable methods for screening COVID 19 infections, even better than PCR testing.

Authors: traditionally, the diagnostic performance of a test is commonly evaluated against a reference test, which is assumed to be a "gold standard" with perfect sensitivity (Se) and specificity (Sp). In our Bayesian model, we carried out a latent class analysis which allows the estimation of Se and Sp of the evaluated diagnostic tests, in a population where the underlying true infection status is unknown according to Hui and Walter, 1980 [Hui, S.L. and Walter, S.D. (1980) Estimating the error rates of diagnostic tests. *Biometrics* 36, 167–71]. More background is provided in the introduction of the manuscript.

2. Reviewer 1: A proof of concept that dogs are able to identify COVID 19 in saliva and Tracheobronchial smears were published by Jendry et al. (Jendry et al. *BMC Infectious Diseases* (2020) 20:536. <https://doi.org/10.1186/s12879-020-05281-3>). In this study sweat was a novel specimen. For the scientific community of medical detection dogs, major information on training, dogs, and testing are missing.

Authors: A supplementary material section is now added describing the details of dogs training and imprinting, including the reward system and the continuous maintenance of the dogs interest. Apart from the known positive and negative specimens used during the training, the test runs included samples that were positive and negatives to represent internal control within the runs.

3. Reviewer 1: It is of utter importance to critically evaluate the odor samples that are presented to the dogs, to judge, what the dogs possibly identify.

Authors: Thanks for the comment. The information has been added on page 2, lines 44-50 and page 10, lines 311-318 in the revised manuscript.

4. Reviewer 1: The author found a very high specificity for the detection dogs. For a screening test for Covid 19 a high predictive value. With the relative low prevalence of the infection, the NPV is nearly perfect.

So if the authors could describe the dog work in more detail, it would be great to spread this information.

Authors: Thanks for the comment. The detailed dog training procedures is added as Supplementary materials and methods.

Reviewer #2 (Remarks to the Author):

1. Reviewer 2: The study is important and timely. However in its current form the manuscript does not allow the reviewer to fully appraise the scientific rigor of the study nor the reader to fully understand the conclusions reached.

Authors: The revised and updated manuscript is now providing additional information on the background, the rationale behind the testing, the training of the dogs, the double-blind aspect of sample collection in addition to the data from 3290 participants included in the study versus the previous figure of 884 participants.

2. Reviewer 2: From the abstract it is not clear how the paper can conclude the dog is more successful than the PCR screening. As it is unclear what gold standard the two are being compared to? This is an issue throughout for me as the first analysis assumes the PCR is the gold standard, yet the Bayesian analysis concludes otherwise, which seems a cyclical argument and is not coherently dealt with in the manuscript

Authors:

In the absence of a reasonable reference test or true gold standard for detection of CNS with known Se and Sp, Bayesian LCA provides an invaluable option for the estimation of Se and Sp of two or more tests without any assumption about the underlying true disease status of each subject according to Hui and Walter, 1980, [Hui, S.L. and Walter, S.D. (1980) Estimating the error rates of diagnostic tests. *Biometrics* 36, 167–71].

Moreover, test validation studies assuming perfect reference tests are common, but with a potential to introduce bias in estimation of index test(s) performances. The true infection status in latent class analysis is regarded as an existing, but unknown (latent) variable, and test accuracy and prevalence are subsequently parametrized according to this latent variable. Therefore, the application of an appropriate statistical framework allowing the estimation of the diagnostic performances in a 'gold-standard independent fashion' is essential.

Here in this work, we conducted both of the two approaches for evaluation of K9 and PCR using the conventional gold standard method and Bayesian latent class analysis to show that the difference between in the diagnostic performance of the two test is influenced by the method of diagnostic test evaluation.

3. Reviewer 2: For example..

Line 121.. in the paragraph above the authors describe that they use BLCA as the two test may produce different results and we cannot make assumptions which represents the true disease state yet here they present TP, TN FP etc.. How can these be derived, if the true gold standard is unknown? Similarly section 150 – 168 all assume the PCR is the gold standard? This discrepancy between the two types of analysis is implicit throughout the paper and requires more synthesis and clearer discussion.

Authors: it is important to clarify that in this research, we are presenting two different approaches for diagnostic test evaluation and it seems this may need some clarification in the text. In conventional reference test, PCR was considered as the gold standard for estimation the performance of K9 dogs. On the other hand, we used carried out a Bayesian latent class analysis for the PCR and K9 without assumption of any of them as a gold standard in two populations.

4. Reviewer 2: The background section is extremely short and goes from describing the multiplicity of uses of dogs to stating the specificity of PCR for Covid-19. It requires some introduction to Covid-19, the needs for screening and references to the sensitivity of PCR for detecting COVID-19 from different sample types.

Authors: Thanks for the comment. The background section has been revised and re-written as per suggestion of respected reviewer.

5. Reviewer 2: The sections on training and testing of the dogs give inadequate detail to allow full appraisal of the study. Was training blind? Was testing blind or double blind? Who laid out the samples? Who handed the dogs? How many dogs were they? What breeds were they? How many times was each sample presented? What was the ratio of positive to negatives presented to the dogs? How were the dogs rewarded? There are sound examples of the sort of details required when reporting dog detection studies in References 9-13 in the current paper, and also studies describing the multiple pitfalls that need to be avoided when training dogs (e.g. Edwards et al 2017; Guest et al 2020)

Edwards, T.L., Browne, C.M., Schoon, A., Cox, C., Poling, A. (2017). Animal Olfactory Detection of Human Diseases: Guidelines and Systematic Review, Journal of Veterinary Behavior: Clinical Applications doi: 10.1016/j.jveb.2017.05.002

Guest CM, Harris R, Anjum I, Concha AR and Rooney NJ (2020) A Lesson in Standardization – Subtle Aspects of the Processing of Samples Can Greatly Affect Dogs' Learning. Front. Vet. Sci. 7:525. doi: 10.3389/fvets.2020.00525

Authors: Thanks for the comment. The detailed procedure for the dogs training addressing the issues raised by the respected reviewers is included in the manuscript as supplementary material and methods.

6. Reviewer 2: Lines 71... was processing of the positive and negative samples the same? What happened if a patient showed no symptoms and tested positive or showed symptoms and tested negative? Were they eliminated from both groups?

Authors:

At the time of the sweat sample collection from the donor, no information was available regarding their PCR test result. Essentially, the sweat specimen and the PCR specimen are collected, by different teams, during the person's visit to the COVID-19 screening center. While the result of the olfaction test is obtained within an hour or so, the PCR test result does not appear until the following day of the specimen collection. The teams processing the specimens have no information about the donors' symptomatology, if any.

7. Reviewer 2: Line 78 should read arm pit not bit

✓ **Authors:** Fixed

8. Reviewer 2: Line 81 should read non-infectious not infective

✓ **Authors:** Done

9. Reviewer 2: Line 92 there appears to be text missing as a sentence finishes mid phrase.

✓ **Authors:** Fixed

10. Reviewer 2: In line 115 the authors describe this to be the first application of BLCA to Covid -19 dog detection data.. But I question, with only 8 positive cases according to PCR, is this an appropriate sample and analysis technique. Has the analysis method been used on other models of canine detection and if so to what benefit? What's more the authors needs to expand the discussion to explain what this analysis brings to the debate and describe fully whether their Bayesian or more conventional NPV and PPV values re likely more accurate.

Authors: as we mentioned in the manuscript, the total study population was 729 individuals and this number was included in the study in a random method during the study period. This sample size is indeed very good for carrying out a robust statistical modelling. It may worth to refer that the Se and Sp estimates of the tests under evaluation is calculated based on all the possible crosstabulation combinations (++, +-, -, --) not only the number of positive cases as in case of gold standard method. The Bayesian latent class analysis has been implemented recently in the veterinary field for estimation of the performance of the available conventional and modern diagnostics techniques for many infectious diseases in different animal species including dogs. Here are a few examples:

- In dogs, Basurco et al., 2020. Evaluation of the performance of three serological tests for diagnosis of *Leishmania infantum* infection in dogs using latent class analysis. *Rev Bras Parasitol Vet.* 2020 Dec 4;29(4):e018020.

- Uiterwijk et al., 2018. Comparing four diagnostic tests for *Giardia duodenalis* in dogs using latent class analysis. *Parasit Vectors.* 2018 Jul 31;11(1):439

- Hartnack et al., 2013. Latent-class methods to evaluate diagnostics tests for *Echinococcus* infections in dogs. *PLoS Negl Trop Dis.* 2013;7(2):e2068.

LCA in other species:

- Elsohaby et al., 2020. *Prev Vet Med.* Aug;181:105054

- Franzo et al., 2019. *Lett Appl Microbiol.* Dec;69(6):417-423

- Mahmmod et al., 2019. *J Appl Microbiol.* Aug;127(2):406-417.

- Svenesen et al., 2018. *Prev Vet Med.* 2018 Dec 1;161:69-74.

- Mahmmod et al., 2013. *Prev Vet Med.* 2013 Nov 1;112(3-4):309-17.

Thanks for your suggestions, more details are added to the discussion to show the importance of Latent class analysis and its superior advantages over traditional methods.

11. Reviewer 2: Table 3 presents analysis which have not been introduced nor fully described in the methods

Authors: Now it is fully described in the manuscript text; page 5, lines 183-187; page 7, lines 230-238 and Figure 2.

12. Reviewer 2: The logic of splitting the sample in to Asian and African is unclear, and since there are no positive cases in the African subpopulation, it seems this adds little to the paper

Authors: As we showed in the manuscript, for running the model, we should fulfill three key assumptions, according to the paradigm described by Hui and Walter (1980); (i) the target population should consist of two or more subpopulations with differing prevalences, (ii) there should be a constant Se and Sp of the index tests across the subpopulations, and (iii) the tests under evaluation should be conditionally independent given the disease status. In our data, we used the geographical location as a stratifier to make two subpopulations for running our model.

13. Reviewer 2: Reference 10 the formatting is incorrect

✓ **Authors:** Fixed

Reviewers' Comments:

Reviewer #1:

Remarks to the Author:

Dear authors,

your revised manuscript has improved vastly. I congratulate to this work. You added valuable data and I can suggest this manuscript now for publication with minor revisions.

In the supplementary section, I am interested if and how the dogs were rewarded in the test scenario. You described that you used "refreshing" samples, but I always wonder if you teach dogs to ignore a negative sample by not rewarding it, what does a dog learn in testing when it is not rewarded for a true positive? I admit this is not an easy question.

Furthermore you stated, that you tested every line up with two dogs.

So did all dogs tested all samples, or every dog 50 % , and how would you judge a sample that was tested by one dog positive and negative by the next one.

For the prevalence of Covid in your study population I come up with 0.005% (18/3290)

Then I found some minor mistakes in the references (9: Edwards!, And a Missing year with 2:Williams)

Reviewer #2:

Remarks to the Author:

Many thanks for your revision. I find the MS much improved. The distinction between the two types of analysis is now much clearer, and the training details help. However, I think there are still important missing details on the dog testing phase and on the sample storage. When these are added I feel the paper will be a valuable addition to the literature.

Minor points

Line 27 "able to successfully screen out, from a cohort of 3290 individuals, persons who were negative for the SARS-CoV" – please add the number of people who were screened out ie those that were negative

Line 45 spelling of bacteria needs correcting

Line 52 should read "a pandemic"

Line 69 FDA acronym needs definition on first time of use

Line 75 what is the gold standard for measuring viral proteins or RNA molecules in a specimen, against which the test are assessed

Line 102 "many reported bias" is grammatically incorrect – replace with "much reporting bias" or "many reporting biases"

Line 102 I don't understand the sentence "The diagnostic test characteristics of real-time PCR and detection dogs test have never been evaluated." do you mean "the diagnostic capability of PCR and dogs have never been compared"?

Line 107 should read "the human population"

Line 126 should read "sample collection" -remove s

Line 134 Were these control samples from patients at the same hospital This is not clear , and confusion is added as the section above already described migrant workers. If they were at the hospital, what were they admitted for?

Line 145 needs a few more details eg trained to search all XX cones and sit in front of the one containing a positive sample whilst ignoring all others. This was trained via positive reinforcement using food or play rewards.

Line 150 should read 3290-axilla sweat samples as armpit is mentioned later; also missing space ahead of individual

Line 155 as per which guidelines. Please specify? Do you mean those in the suppl material, if so state who they are prepared by

Line 193 It's not clear which two populations the authors are referring to – do they mean the training and the testing population described in 2.1, and 2,2 respectively?) Later you talk about Asian and African but its not cellar here

Line 218 when referring to sens and spec – can you include the equations using the codes as included in Table 1

Line 246 What do you authors mean by cleaning – what cleaning was conducted? Should read "subjects not being from Africa or Asia"

Line 295 What gold standard are these studies using when calculating sensitivity and specific

Line 296 should read "requirement fulfilment of"

The end of paper would benefit from some discussion of how to subsequently test / confirm if the results of Bayesian modelling are meaningful in the real world

I think more information on dog testing protocol is needed in the main paper also details of how sample were stored.. were they frozen? And how long were they stored for?

Supplementary training material i-s very useful, but it need s a little work and proof reading as the tenses are currently inconsistent

Page 1 Dog's smell different than hum – needs correcting do you mean "dogs smell differently"?

This sentence seems to be conjecture – how can we know this burger analogy?

Variable reward schedules - the definition seems incorrect. This usually refers to a non constant ratio of reward to non reward for positive responses

Page 6 should read "footwear"

Page 9 "where the results showed positive to start operations."-I do not follow this sentence – please can it be clarified

"All participants signed a consent form, prior to providing the sweat samples"- does not belong in this training protocol. Its mentioned in eth main MS I

During testing it is unclear if all dogs searched the same samples line-ups or not nor whether the sample samples were searched by multiple dogs- =

There still remain very few details on the dog testing protocol. I believe these belongs in the main manuscript. Who was present during testing? How long was the session? How were hesitations coded? Were they filmed? Who decided if the dog indicated or not? Did dogs always search all stands even after finding appositve sample? Were the same samples only used once? Or were they presented to multiple dogs?

Response to Referees' Comments

No	Reviewers' Comments	Author's Response to Reviewers' Comments
	Reviewer #1 (Remarks to the Author):	
1	Your revised manuscript has improved vastly. I congratulate to this work. You added valuable data and I can suggest this manuscript now for publication with minor revisions.	Thank you for the valuable suggestion
2	In the supplementary section, I am interested if and how the dogs were rewarded in the test scenario. You described that you used "refreshing" samples, but I always wonder if you teach dog to ignore a negative sample by not rewarding it, what does a dog learn in testing when it is not rewarded for a true positive? I admit this is not an easy question.	In the testing phase (double-blind) no one knows on the spot if it is a true positive. So we do not reward with a toy, but we do reward with a praise. This is also the reason why we randomly reward in the final stages of the training phase, varying between toy and/or praise. We use in between the test "refreshing samples" (we call those calibration samples) to keep the dog motivated and in this case we do reward on an indication on a true positive.
3	Furthermore you stated, that you tested every line up with two dogs. So did all dogs tested all samples, or every dog 50 % , and how would you judge a sample that was test by one dog positive and negative by the next one.	Please refer to the Supplement under section 3.3 (Operational Training and Testing)
4	For the prevalence of COVID-19 in your study population I come up with 0.005% (18/3290)	Yes, this is now reflected in Table 2 of the manuscript, Line 265.
5	Then I found some minor mistakes in the references (9: Edwards!, And a Missing year with 2:Williams)	Minor mistakes in the references are fixed.
	Reviewer #2 (Remarks to the Author):	
6	Many thanks for your revision. I find the MS much improved. The distinction between the two types of analysis is now much clearer, and the training details help.	Thank you for the comment
7	However, I think there are still important missing details on the dog testing phase and on	Addressed in Supplement under section 3.3 (Operational Training and Testing)

	the sample storage. When these are added I feel the paper will be a valuable addition to the literature.	
	Minor points	
8	Line 27 “ able to successfully screen out, from a cohort of 3290 individuals, persons who were negative for the SARS-CoV” – please add the number of people who were screened out ie those that were negative	Number of people who were screened out is added as per reviewer's comments. Line 28.
9	Line 45 spelling of bacteria needs correcting	The term "bacteriuria" is used to denote the presence of bacteria in the urine. Line 46.
10	Line 52 should read “a pandemic”	Done. Line 53.
11	Line 69 FDA acronym needs definition on first time of use	FDA acronym is defined. Line 71.
12	Line 75 what is the gold standard for measuring viral proteins or RNA molecules in a specimen, against which the test are assessed	RT-PCR is the most widely used test and employed as the gold standard when assessing the K9 test. Line 75.
13	Line 102 “many reported bias” is grammatically incorrect – replace with “!much reporting bias” or “many reporting biases”	Done. Line 104.
14	Line 102 I don't understand the sentence “The diagnostic test characteristics of real-time PCR and detection dogs test have never been evaluated.” do you mean “the diagnostic capability of PCR and dogs have never been compared”?	The sentence is rephrased. Line 104.
15	Line 107 should read “the human population”	Done. Line 109.
16	Line 126 should read “sample collection” -remove s	Done. Line 128.
17	Line 134 Were these control samples from patients at the same hospital This is not clear , and confusion is added as the section above already described migrant workers. If they were at the hospital, what were they admitted for?	Yes, the negative samples also came from the same hospital. The hospital staff (doctors, nurses and others) and other patients provided negative samples. Criteria was to have a minimum of 2 consecutive negative PCR tests and no signs or symptoms of respiratory tract infection, cold or flu. Due to privacy reasons, we did not receive full information on the patients' exact diagnosis on admission. Line 135.

18	Line 145 needs a few more details eg trained to search all XX cones and sit in front of the one containing a positive sample whilst ignoring all others. This was trained via positive reinforcement using food or play rewards.	More details are provided in the manuscript in section 3.0 training of dogs. Line 167.
19	Line 150 should read 3290- armpit sweat samples as armpit is mentioned later; also missing space ahead of individual	Corrected. Line 152.
20	Line 155 as per which guidelines. Please specify? Do you mean those in the supplementary material, if so state who they are prepared by	Line 174. Dutch Animal welfare regulations- https://www.government.nl/topics/animal-welfare/welfare-of-pets
21	Line 193 It's not clear which two populations the authors are referring to – do they mean the training and the testing population described in 2.1, and 2,2 respectively?) Later you talk about Asian and African but its not cellar here	This is now clarified as African and Asian populations. Line 228.
22	Line 218 when referring to sens and spec – can you include the equations using the codes as included in Table 1	Done. Table 1. Line 254.
23	Line 246 What do you authors mean by cleaning – what cleaning was conducted? Should read “subjects not being from Africa or Asia”	We mean that data were initially screened for any missing information or values before conducting any meaningful analysis. Based on that, out of the total of 3290 observations, 156 observations were excluded from the analysis due to missing information or subjects not being from Africa or Asia. The analysis was carried out on the clean data with complete observations that included 3134 observations. This clarification has been added to the text (Line 286 - 288). The exclusion of the 156 observations had negligible effect on the estimates of the sensitivity and specificity based on gold standard method.
24	Line 295 What gold standard are these studies using when calculating sensitivity and specific	Different gold standards were used. Ren et al. (2020) compared the CT scan and RT-PCR using McNemar correlation test. Wang et al. (2020) used clinical indicators based on symptoms and radiology as a gold standard for calculation of Se and Sp of RT-PCR. Yu et al. (2020) used droplet digital PCR (ddPCR) as a gold standard.
25	Line 296 should read “requirement fulfillment of”	Done. Line 337.

26	The end of paper would benefit from some discussion of how to subsequently test / confirm if the results of Bayesian modelling are meaningful in the real world	Done. A short paragraph on the raised issue on Bayesian modelling has been added to the text. (Line 360-367).
27	I think more information on dog testing protocol is needed in the main paper also details of how sample were stored.. were they frozen? And how long were they stored for?	The following statement is copied from the Supplementary document. Samples that were collected from the hospital were transported on the same day to the Federal Customs K9 facility. The positive and negative samples were transported separately in medical coolers with ice packs, but under the same conditions. From there, they were stored in the fridge and mostly used the same day. All samples were taken out of the fridge to come to 'room temperature' before presented to the dogs. If samples were used a day later, they were stored in an air tight container in the fridge. The operational samples were collected by the participant, under supervision of an assigned nurse or medical doctor explaining the procedure and making sure that the sampling protocol was followed. The samples were stored in airtight containers. Lines 157-165.
28	Supplementary training material is very useful, but it needs a little work and proof reading as the tenses are currently inconsistent	Done
29	Page 1 Dog's smell different than hum – needs correcting do you mean “dogs smell differently”? This sentence seems to be conjecture – how can we know this burger analogy?	Please see amendment in supplement; related to the burger; dogs have smell receptors that are 10,000 times more accurate than human and besides that, the olfactory system in the dog's brain, dedicated to processing smells, is much larger in dog's compared to humans. This allows the dogs to distinguish and remember a wide variety of scents.
30	Variable reward schedules - the definition seems incorrect. This usually refers to a non-constant ratio of reward to non-reward for positive responses	We used the variable ratio reinforcement schedule- see amendment in supplement
31	Page 6 should read “footwear”	Done in supplement.
32	Page 9 “where the results showed positive to start operations.”-I do not follow this sentence – please can it be clarified	See amendment in the supplement.
33	“All participants signed a consent form, prior to providing the sweat samples”- does not belong in this training protocol. Its mentioned in eth main MS I	Removed from supplement as per suggestion

34	During testing it is unclear if all dogs searched the same samples line-ups or not nor whether the sample samples were searched by multiple dogs.	See amendment in the supplementary document under section 3.3 (Operational Training & Testing)
35	There still remain very few details on the dog testing protocol. I believe these belongs in the main manuscript. Who was present during testing? How long was the session? How were hesitations coded? Were they filmed? Who decided if the dog indicated or not? Did dogs always search all stands even after finding appositve sample? Were the same samples only used once? Or were they presented to multiple dogs?	1. Q- who was present during testing A) 1 Trainer- 1 data registrar - 1 team leader, sometime other instructors or the authors of the paper (Line 189-192) 2. Q How long was the session? A) Full session including placing six samples, registration, search by 2 dogs and removing six samples & cleaning = around 4 to 10 minutes depending on number of dogs. 3. Q- How were the hesitations coded: A) If only 1 dog out of 4 dogs gave a hesitation, this was recorded as negative, if 2 or more dogs gave a positive indication or hesitation this was recorded as positive. 4. Q- were they filmed; A) No, not contsantly, but several stages in the process were filmed. 5. Q- Who decided if the dog indicated? A) The team leader or instructor decided if this was an indication or not. 6. Q- Did the dog always search all stands A) Yes, the dogs always screened all stands even after finding positive sample. 7. Q- were the samples used once A) No, the operational samples were screened by at least 2 dogs. The training samples were used by multiple dogs. 8. Q- or were they presented to multiple dog A) Yes, they were presented and screened by at least 2 dogs. Please refer to the Supplement under section 3.3 (Operational Training and Testing).